# REVISITING OVER-SMOOTHING IN GRAPH NEURAL NETWORKS

## ABSTRACT

Shallow graph neural networks (GNNs) are state-of-the-art models for relational data. However, it is known that deep GNNs suffer from over-smoothing where, as the number of layers increases, node representations become nearly indistinguishable and model performance on the downstream task degrades significantly. Despite multiple approaches being proposed to address this problem, it is unclear when any of these methods (or their combination) works best and how they perform when evaluated under exactly the same experimental setting. In this paper we systematically and carefully evaluate different methods for addressing over-smoothing in GNNs. Furthermore, inspired by standard deeply supervised nets, we propose a general architecture that deals with over-smoothing based on the idea of layer-wise supervision. We term this architecture deeply supervised GNNs (or DSGNNs for short). Our experiments show that deeper GNNs can indeed provide better performance when trained on a combination of different approaches and that DSGNNs are robust under various conditions and can provide best performance in missing features scenarios.

## 1 INTRODUCTION

Graph Neural Networks, first introduced by Scarselli et al. (2009), have emerged as the de facto standard for representation learning on relational or graph-structured data. GNNs find many useful applications in building predictive models in traffic speed prediction, product recommendation, and drug discovery (Zhou et al., 2020a; Gaudelet et al., 2021). One of the most important applications of GNNs is that of *node property prediction*, as in semi-supervised classification of papers (nodes) in a citation network (see, e.g., Kipf & Welling, 2017). In this case, we are given labels for a subset of the nodes and aim to learn an algorithm that can accurately predict the labels for the remaining nodes using the network structure and (if available) node features.

Even though GNNs, and especially those based on the Graph Convolutional Network (GCN) formulation (Kipf & Welling, 2017) and its extensions (Veličković et al., 2018; Hamilton et al., 2017; Wu et al., 2019; Klicpera et al., 2018), have been shown to be a powerful tool for graph representation learning, they are limited in depth, that is the number of graph convolutional layers. Indeed, deep GNNs suffer from the problem of over-smoothing where, as the number of layers increases, the node representations become nearly indistinguishable and model performance on the downstream task deteriorates significantly. Increasing model depth is necessary in order to allow information to travel between distant nodes in the graph where each graph convolutional layer corresponds to propagating information from a node's one-hop neighbourhood.

Previous work has analyzed and quantified the over-smoothing problem (Liu et al., 2020; Zhao & Akoglu, 2020; Chen et al., 2020a) as well as proposed methodologies to address it explicitly (Li et al., 2018; Zhao & Akoglu, 2020; Xu et al., 2018). Some of the most recent approaches have focused on forcing diversity on latent node representations via residual connections (see, e.g, Xu et al., 2018; Chen et al., 2020b), normalization (see, e.g, Zhou et al., 2020b; Zhao & Akoglu, 2020), and enforced sparsity (see, e.g, Rong et al., 2020). However, the proposed solutions have mostly been shown to *alleviate* over-smoothing but do not completely eliminate it, with shallow networks usually performing best. In this context, alleviating means that performance does not catastrophically deteriorate as a function of network depth. One notable exception to this is the

GCNII architecture (Chen et al., 2020b), which was shown to improve the performance of standard GNNs on classification tasks when using deeper architectures.

One interesting scenario for the analysis of deep GNNs is when only a subset of the nodes have features, which we will refer to as the missing-feature setting (Zhao & Akoglu, 2020). In this case, most of the solutions mentioned above (with the exception of GCNII) have been shown superior to the basic GCN architecture. Intuitively, as pointed out by Zhao & Akoglu (2020), a large number of propagation steps (i.e., deeper GNNs) may be required to obtain useful feature node representations.

While acknowledging the significant advances towards making GNNs more robust to over-smoothing, we have found important gaps in the literature that, we believe, need to be recognized and addressed by the community. Firstly, and naturally, most previous work has focused on developing new algorithms and showing that they outperformed previous approaches. While there is nothing inherently wrong with such approaches, this usually has come at the expense of empirical results across different algorithms not using exactly the same settings (such as hyper-parameter optimization). Secondly, we have also found that, in fact, all proposed solutions are general enough that can be combined together. However, these combinations and their performance have not been studied in detail. Finally, another crucial gap in the GNN over-smoothing literature is that, previous approaches that have tackled (seemingly different but) related problems in standard neural networks have not been investigated. In particular, we refer to the work on deeply supervised nets Lee et al. (2015) for learning discriminative and robust features and for dealing with vanishing/exploding gradients.

**Contributions**: In this paper, (i) we address the above gaps by systematically and carefully evaluating several proposed GNN over-smoothing solutions and their combinations. We analyze their performance in the transductive, semi-supervised node classification setting in both the standard fully observed and missing-feature settings. Furthermore, inspired by the work of Lee et al. (2015), (ii) we propose a new general architecture for tackling over-smoothing. Our architecture trains predictors using node representations from all layers, each contributing to the loss function equally, therefore encouraging the GNN to learn discriminative features at all network depths. We name our approach deeply-supervised graph neural networks (DSGNNs). (iii) We show that DSGNNs are resilient to the over-smoothing problem in deep networks and can outperform competing methods on challenging datasets. Finally, (iv) we provide recommendations for the selection of a GNN architecture for practical applications.

## 2 GRAPH NEURAL NETWORKS

Let a graph be represented as the tuple $G = (V, E)$ where $V$ is the set of nodes and $E$ the set of edges. The graph has $|V| = N$ nodes. We assume that each node $v \in V$ is also associated with an attribute vector $\mathbf{x}_v \in \mathbb{R}^d$ and let $\mathbf{X} \in \mathbb{R}^{N \times d}$ represent the attribute vectors for all nodes in the graph. Let $\mathbf{A} \in \mathbb{R}^{N \times N}$ represent the graph adjacency matrix; here we assume that $\mathbf{A}$ is a symmetric and binary matrix such that $\mathbf{A}_{ij} \in \{0, 1\}$, where $\mathbf{A}_{ij} = 1$ if there is an edge between nodes $i$ and $j$, i.e., $(v_i, v_j) \in E$, and $A_{ij} = 0$ otherwise. Also, let $\mathbf{D}$ represent the diagonal degree matrix such that $\mathbf{D}_{ii} = \sum_{j=0}^{N-1} \mathbf{A}_{ij}$.

Typical GNNs learn node representations via a neighborhood aggregation function. Assuming a GNN with $K$ layers, we define such a neighborhood aggregation function centred on node $v$ at layer $l$ as follows,

$$\mathbf{h}_v^{(l)} = h^{(l)} \left( f \left( g \left( \mathbf{h}_v^{(l-1)}, \mathbf{h}_u^{(l-1)} \, \forall u \in \mathcal{N}_v \right) \right) \right), \tag{1}$$

where $\mathcal{N}_v$ is the set of node $v$'s neighbors in the graph, $g$ is an aggregation function, $f$ is a linear transformation that could be the identity function, and $h^{(l)}$ is a non-linear function applied element-wise. Let $\mathbf{H}^{(l)} \in \mathbb{R}^{N \times d^{(l)}}$ the representations for all nodes at the $l$-th layer with output dimension $d^{(l)}$; we set $\mathbf{H}^{(0)} \stackrel{\text{def}}{=} \mathbf{X}$ and $d^{(0)} \stackrel{\text{def}}{=} d$. A common aggregation function $g$ that calculates the weighted average of the node features where the weights are a deterministic function of the node degrees is $\hat{\mathbf{A}}\mathbf{H}$ as proposed by Kipf & Welling (2017). Here $\hat{\mathbf{A}}$ represents the twice normalized adjacency matrix with self loops given by $\hat{\mathbf{A}} = \hat{\mathbf{D}}^{-1/2}(\mathbf{A} + \mathbf{I})\hat{\mathbf{D}}^{-1/2}$ where $\hat{\mathbf{D}}$ is the degree matrix for $\mathbf{A} + \mathbf{I}$ and $\mathbf{I} \in \mathbb{R}^{N \times N}$ is the identity matrix. Substituting this aggregation function in 1 and specifying

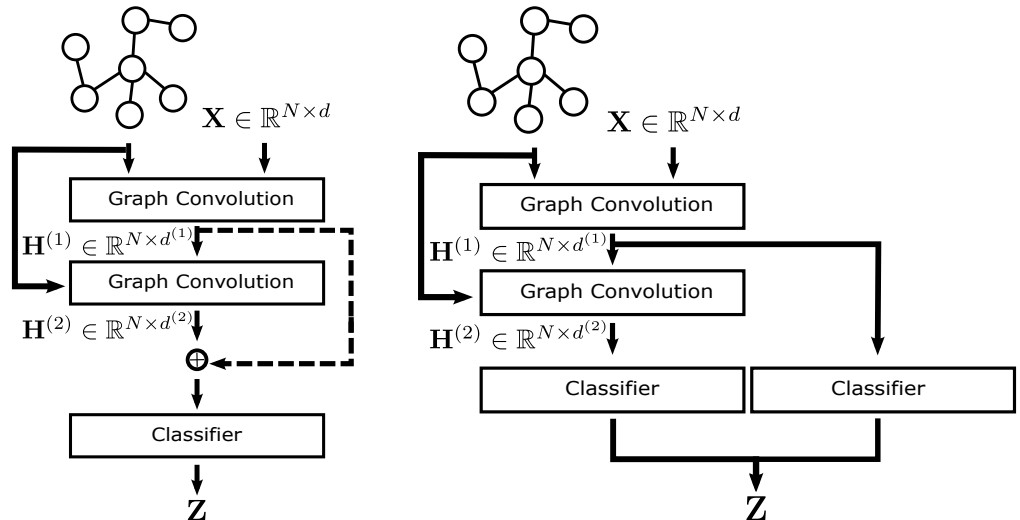

Figure 1: GNN architectures for node classification. *Left*: the standard architecture using two graph convolutional layers but also shown with optional jump connections (dashed lines). *Right*: the proposed architecture with deep supervision.

$f$ to be a linear projection with weights $\mathbf{W}$ , gives rise to the graph convolutional layer of Kipf & Welling (2017),

$$\mathbf{H}^{(l)} = h^{(l)}(\hat{\mathbf{A}}\mathbf{H}^{(l-1)}\mathbf{W}^{(l)}), \tag{2}$$

where, as before, $h^{(l)}$ is a non-linear function, typically the element-wise rectified linear unit (ReLU) activation (Nair & Hinton, 2010).

Many other aggregation functions have been proposed, most notably the sampled mean aggregator in GraphSAGE (Hamilton et al., 2017) and the attention-based weighted mean aggregator in graph attention networks (GAT, Veličković et al., 2018).

## 2.1 Node Property Prediction

Equation 2 is a realization of 1 and constitutes the so-called spatial graph convolutional layer. More than one such layers can be stacked together to define GNNs. When paired with a task-specific loss function, these GNNs can be used to learn node representations in a semi-supervised setting using full-batch gradient descent. For example, in semi-supervised node classification, it is customary to use the row-wise softmax function at the output layer along with the cross-entropy loss over the training (labeled) nodes.

Figure 1 (left) shows a diagram of the standard GNN architecture with optional jump connections as proposed in Xu et al. (2018). Furthermore, the last hidden layer can be followed by a multi-layer Perceptron (MLP) that functions as the classifier. The MLP is optional when using the standard GCN architecture (Kipf & Welling, 2017) but necessary when employing jump connections. Given a suitable loss function such as the cross-entropy for classification, we can train predictive models in a semi-supervised setting for node-level tasks.

## 3 Over-Smoothing in Graph Neural Networks

Intuitively, deeper networks should allow information to travel between distant nodes in the graph and we would like to learn node representations that take into account a larger neighborhood of a node. In other words, nodes further away in the graph should influence the node's representation and take advantage of the graph's structure beyond the local neighborhoods. However, in practice, this is not the case due to over-smoothing. As mentioned in section 1, over-smoothing in GNNs is

the phenomenon where model performance on the downstream task deteriorates significantly as the number of layers increases. This can be attributed to the repeated application of the neighborhood aggregation operation and, therefore, node representations becoming nearly indistinguishable for very deep architectures.

## 3.1 PREVIOUS APPROACHES

We briefly overview some of the techniques proposed in the literature to address smoothing in GNNs, which we will use in our experimental evaluation.

**Graph convolutional networks (GCNs)**: Kipf & Welling (2017) identified over-smoothing in deep GNNs when using GCN layers, showing empirically that such architectures performed best with only 2 or 3 layers, while performance degraded substantially for 7 or more layers. They associated this problem with larger node-context sizes and overfitting, due to the significant increase in model parameters. They proposed the addition of residual connections for each layer such that the input node features where combined with the layer's node output features. Residual connections alleviated the over-smoothing problem but the best performance was still achieved using only 2 or 3 layers.

**Jumping knowledge networks (JKNets)**: Xu et al. (2018)'s JKNets were specifically proposed as a solution to over-smoothing and centered on the introduction of residual connections. JKNet introduces residual connections from all hidden layers to the output layer. Node attributes are combined using one of several schemes, e.g., concatenation and element-wise max-pooling, to generate node representations adapted independently to different context sizes for each node. JKNet was shown empirically to alleviate over-smoothing but best performance was limited to shallow networks.

**Pair normalization (PN)**: Zhao & Akoglu (2020) provided a first attempt at quantifying over-smoothing. They proposed 2 metrics that measure the pairwise similarity of latent node representations, *row-diff*, and the variance of latent node features, *col-dif*, across all nodes in the graph. The authors propose that the node representations output at each hidden layer should preserve the total pairwise squared distance (TPSD) of the input node attributes. They propose a pair normalization (PN) layer that aims at keeping the TPSD constant across each hidden layer in the GNN. They demonstrate empirically that equipping a GNN with PN layers alleviates over-smoothing but does not improve performance over shallow networks without layer normalization. In fact, adding pair normalization layers to a shallow GNN is shown to decrease performance. However, they introduce the missing-feature setting where layer normalization is shown to improve model accuracy as a function of increased depth. In this setting, only the attributes for the nodes in the training set are known whereas the attributes for the nodes in the test set are unknown. We note that, in this setting, model performance is lower than fully-observed setting.

**Group normalization (GN)**: Generalizing the pair normalization idea, differentiable group normalization (Zhou et al., 2020b) introduces a constraint that attempts to force latent node representations into groups such that nodes with similar labels cluster together whereas groups of nodes with different labels are well separated. Such grouping of latent representations should benefit the performance of a downstream classifier. GN alleviates over-smoothing as network depth increases but does not help improve performance over shallow models. However, on the missing-feature setting, GN was shown to outperform pair normalization and benefit from added network depth.

**GCNII**: The GCNII architecture, as proposed by Chen et al. (2020b), extends GCN in two ways. First, it modifies the residual connections to carry information from the input features to each hidden layer rather than the latent representations of the previous hidden layer. Second, it introduces an identity mapping to the hidden layer weights as initially proposed by He et al. (2016). The latter guarantees that GCNII will perform as well as an equivalent (in terms of number and size of hidden layers) GCN model. GCNII is the only architecture shown empirically to benefit from network depth. However, this architecture has not been evaluated in the missing-feature setting where it is not clear if the lack of input features will limit its performance.

Unavoidably, there are other techniques for dealing with over-smoothing in GNNs that we do not investigate here. Of notable mention is the work of Rong et al. (2020), who proposed DropEdge as a general heuristic for alleviating over-smoothing by modifying the message passing mechanism in GNNs. The core idea is to prevent some of the information from propagating through the graph by randomly eliminating a subset of the graph's edges. At each training epoch, a random subset of the

graph's edges are removed preventing information flow through them; these edges are restored for the next epoch before another subset of edges is removed. They show that DropEdge is effective at alleviating over-smoothing and that a small performance boost is possible for shallow GCN models augmented with DropEdge. However, they do not evaluate DropEdge in the low-label regime that is the most common evaluation setting in the GNN literature.

## 3.2 POTENTIAL PITFALLS OF PREVIOUS APPROACHES

One common characteristic of all these previous approaches to over-smoothing in GNNs is that they have not been evaluated using a common framework. For example, some methods are evaluated in the fully-observed setting using random splits whereas others have been evaluated in the low label regime using the node splits published by Yang et al. (2016). In addition, few of these works have been evaluated on both the fully-observed and the missing-feature settings. Lastly, combinations of the different methods have not been studied empirically and, for example, it is unclear if adding layer normalization to GCNII will further improve or hinder downstream task performance.

It is certainly not our intention to focus on the potential flaws in previous approaches to over-smoothing in GNNs. We recognize machine learning is heavily driven by practical performance and that the methods mentioned above have provided significant advances to the field. However, it is imperative that we evaluate our methods fairly and provide advice to practitioners and researchers on best practices and what works best under different scenarios. In fact, the difficulty of consistently evaluating GNNs has been considered previously in works such as Shchur et al. (2018) and Errica et al. (2019).

Despite the issues about fairness in evaluating GNNs as raised in the latter works, inconsistency in reporting GNN performance has persisted. For example, when Chen et al. (2020b) compares with prior works the performance values as reported in Fey & Lenssen (2019) and Rong et al. (2020) are re-used. As a second example, Zhao & Akoglu (2020) uses 32 hidden units for each layer, Zhou et al. (2020b) uses 16 hidden units and Chen et al. (2020b) claims to tune the number of units as a hyper-parameter but the range of values considered is not reported. For these reasons, and in the spirit of fairness, we perform a careful empirical evaluation using the same framework comparing these methods, their combinations and our newly-proposed deeply-supervised graph neural networks (DSGNNs) for both the fully-observed and the missing-feature settings. In the next section we describe the intuition underlying DSGNNs and elaborate on the details on how to train these architectures.

## 4 DEEPLY-SUPERVISED GRAPH NEURAL NETWORKS

Deeply-supervised nets (DSNs, Lee et al., 2015) were proposed as a solution to several problems in training deep neural networks. By using companion objective functions attached to the output of each hidden layer, DSNs tackle the issue of vanishing gradients. Furthermore, in standard neural networks with shallow architectures, deep supervision operates as a regularizer of the loss at the last hidden layer. Lastly, and more importantly, for deep networks, it encourages the estimation of discriminative features at all network layers (Lee et al., 2015). Therefore, inspired by this work, we introduce deeply supervised graph neural networks (DSGNNs), i.e., graph neural network architectures trained with deep supervision. Thus, we hypothesize that DSGNNs are resilient to over-smoothing and test this hypothesis by evaluating and analyzing their performance in training shallow and deep networks in section 5.

Figure 1 (right) shows a diagram of the proposed DSGNN architecture. We introduce jump connections from each hidden layer where node-level representations, as described in section 2, are estimated and given to an MLP that predicts the class probabilities for each node. We note that we use one MLP for each hidden layer as in DSNs. Let $\mathbf{Z}^{(l)} = h^{(l)}(\mathbf{H}^{(l)}\mathbf{W}^{(l)})$ denote the node representations output by a one-layer MLP attached to the $l$-th hidden layer where $h^{(l)}(\cdot)$ is the softmax function for a classification task. Finally, given a loss function, e.g., cross entropy, over our true and predicted outputs, we formulate *layer-dependent losses* and learn all model parameters by optimizing the average loss function across all layers.

## 4.1 Node Classification with a 2-Layer Network

As an illustrative example, here we consider a node classification problem with $C$ classes using 2 GCN layers as shown in figure 1 (right). We are given a graph represented as the tuple $G = (V, E)$ where $V$ is the set of nodes and $E$ the set of edges, as described in section 2. A subset of $M$ nodes, $V_l \subset V$, has known labels. Each label represents one of $C$ classes using a one-hot vector representation such that $\mathbf{Y} \in \mathbb{R}^{M \times C}$. The node property prediction task is to learn a function $f : V \to Y$ that maps node representations to class probabilities.

Consider the case of a 2-layer GNN with GCN (Kipf & Welling, 2017) layers. The node representations output by each of the 2 GCN layers are given by,

$$\mathbf{H}^{(1)} = \text{ReLU}(\hat{\mathbf{A}}\mathbf{X}\mathbf{W}^{(1)}), \tag{3}$$

$$\mathbf{H}^{(2)} = \text{ReLU}(\hat{\mathbf{A}}\mathbf{H}^{(1)}\mathbf{W}^{(2)}), \tag{4}$$

where the ReLU activations are element-wise, $\hat{\mathbf{A}}$ are the edge weights given by equation 2, and $\mathbf{W}^{(i)}$ are trainable layer weights.

Let each GCN layer be followed by a linear layer with softmax activation calculating class probabilities for all nodes in the graph such that,

$$\mathbf{Z}^{(l)} = \text{softmax}(\mathbf{H}^{(l)}\widehat{\mathbf{W}}^{(l)}), \quad l = 1, 2, \tag{5}$$

where $\mathbf{Z}^{(l)}$ are the class probabilities for all nodes as predicted by the $l$th layer, and $\widehat{\mathbf{W}}^{(l)}$ are the layer's trainable weights.

Now we can compute layer-dependent losses as:

$$\mathcal{L}_N^{(l)} = -\sum_{v \in V_l} \sum_{c=0}^{C-1} \mathbf{Y}_{v,c} log(\mathbf{Z}_{v,c}^{(l)}), \quad l = 1, 2. \tag{6}$$

For a standard GNN, in order to estimate the weights $\{\mathbf{W}^{(1)}, \mathbf{W}^{(2)}, \widehat{\mathbf{W}}^{(2)}\}$, we optimize the cross-entropy loss calculated over the set of nodes with known labels only using $\mathcal{L}_N^{(2)}$.

Deep supervision adds a linear layer corresponding to each GCN layer in the model such that, in our example, the model makes two predictions for each node, $\mathbf{Z}^{(1)}$ and $\mathbf{Z}^{(2)}$. We now estimate the weights $\{\mathbf{W}^{(1)}, \mathbf{W}^{(2)}, \widehat{\mathbf{W}}^{(1)}, \widehat{\mathbf{W}}^{(2)}\}$, and optimize the mean loss given by,

$$\mathcal{L}_N = \frac{1}{L} \sum_{k=1}^{L} \mathcal{L}_N^{(k)}, \tag{7}$$

where, in our example, $L = 2$. We estimate the model parameters using gradient-based optimization so as to minimize the total loss in equation 7. Unlike Lee et al. (2015), we do not decay the contribution of the surrogate losses as a function of the training epoch. Consequently, at prediction time we average the outputs from all classifiers and then apply the softmax function to make a single prediction for each node.

## 4.2 Advantages of Deep Supervision

As mentioned before, over-smoothing leads to node representations with low discriminative power at the last GNN layer. This hinders the deep GNN's ability to perform well on predictive tasks. DSGNNs circumvent this issue as the learned node representations from all hidden layers inform the final decision. The distributed loss encourages node representations learned at all hidden layers to be discriminative such that network predictions do not rely only on the discriminative power of the last layer's representations.

Furthermore, deep supervision increases the number of model parameters linearly to the number of MLP layers. Consider a classification model with $K$ graph convolutional layers, $d_G$ dimensional node representations, and a single layer MLP. If the number of classes is $C$, then a DSGNN model requires $K \times d_G \times C$ parameters more than a standard GNN. This is in sharp contrast with other architectures such as that of Liu et al. (2020), where the number of additional parameters grow quadratically with the number of GNN layers.

Table 1: Mean test accuracy $\pm$ one standard deviation for the **fully observed setting**. The notation +PN and +GN indicates the addition of pair and group normalization layers respectively. In parentheses we indicate the architecture depth that achieved the highest test accuracy.

| Method | Cora | Citeseer | Amazon Photo | Pubmed |
|---|---|---|---|---|
| GCN | 82.3±0.5 (3) | 70.4±0.7 (2) | 90.7±0.5 (2) | 79.8±0.5 (2) |
| JKNet | 81.6±0.6 (2) | 69.2±1.4 (2) | 90.3±0.4 (8) | 79.6±0.6 (2) |
| DSGNN | 82.6±0.6 (3) | 69.9±0.7 (2) | 90.6±0.6 (3) | 78.8±0.6 (4) |
| GCNII | 83.1±0.6 (32) | $\mathbf{72.1 \pm 0.4(32)}$ | $\mathbf{92.0 \pm 0.3(4)}$ | 80.1±0.5 (16) |
| GCN+PN | 78.4±0.7 (2) | 62.3±1.2 (2) | 90.0±0.4 (2) | 77.2±0.6 (2) |
| DSGNN+PN | 77.7±1.0 (16) | 63.1±1.3 (2) | 90.6±0.4 (3) | 76.9±1.4 (32) |
| GCNII+PN | 80.9±1.0 (16) | 62.7±2.2 (16) | 89.4±0.6 (32) | 76.0±1.0 (3) |
| GCN+GN | 83.1±0.7 (2) | 69.3±0.6 (2) | 90.7±0.5 (2) | 79.6±0.7 (3) |
| DSGNN+GN | 82.6±0.6 (3) | 70.4±0.7 (2) | 90.6±0.5 (4) | 79.4±0.6 (4) |
| GCNII+GN | $\mathbf{83.6 \pm 0.7(32)}$ | 71.9±0.5 (32) | $\mathbf{92.0 \pm 0.4(8)}$ | $\mathbf{80.2 \pm 0.3(8)}$ |

Table 2: Mean test accuracy $\pm$ one standard deviation for the **missing features setting**. The notation +PN and +GN indicates the addition of pair and group normalization layers respectively. In parentheses we indicate the architecture depth that achieved the highest test accuracy.

| Method | Cora | Citeseer | Amazon Photo | Pubmed |
|---|---|---|---|---|
| GCN | 68.2±1.2 (4) | 38.8±4.8 (4) | 84.6±1.0 (4) | 49.7±2.0 (4) |
| JKNet | 66.1±1.7 (8) | 36.7±3.0 (4) | 65.4±3.6 (8) | 48.0±2.7 (8) |
| DSGNN | 72.6±1.2 (8) | 38.2±2.8 (8) | 85.8±0.8 (8) | 56.6±9.2 (32) |
| GCNII | 75.7±1.2 (32) | 46.1±3.2 (32) | 60.1±2.8 (4) | 62.3±4.2 (16) |
| GCN+PN | 69.8±4.3 (8) | 39.4±2.8 (8) | 85.3±1.2 (8) | 64.9±4.3 (32) |
| DSGNN+PN | 77.4±0.9 (16) | 50.0±2.3 (32) | $\mathbf{87.9 \pm 1.2(16)}$ | $\mathbf{73.8 \pm 1.2(32)}$ |
| GCNII+PN | 71.0±3.4 (16) | 40.6±2.7 (32) | 48.6±3.4 (3) | 67.5±2.3 (32) |
| GCN+GN | 70.8±1.3 (4) | 42.5±4.2 (4) | 84.4±0.8 (4) | 66.2±3.6 (32) |
| DSGNN+GN | 76.2±2.0 (64) | 48.8±5.0 (64) | 86.5±1.4 (16) | 69.3±2.4 (32) |
| GCNII+GN | $\mathbf{78.3 \pm 1.6(32)}$ | $\mathbf{52.7 \pm 2.3(64)}$ | 85.7±1.0 (8) | 71.9±1.3 (32) |

## 5 EXPERIMENTS

We implemented[1] all architectures using PyTorch and the Deep Graph Library (DGL, Wang et al., 2019). The version of the datasets we use is that available via DGL[2]. All experiments were run on workstations with 32GB of RAM, Nvidia Telsa P100 GPU, and Intel Xeon processor.

We evaluate the performance of all architectures under two settings that we call (a) fully observed, and (b) missing features. In the fully observed setting, the node features for all nodes are available during training. In the missing-feature setting, only the node features for those nodes in the training set are available. For all other nodes, we follow Zhao & Akoglu (2020) and set the node attributes to zero vectors. The missing-feature setting reflects a scenario common in real-world applications where obtaining node attribute data can be challenging.

**Datasets:** We use 4 benchmark datasets common in the GNN literature. These are the citation networks Cora, Citeseer, Pubmed and the co-purchase network Amazon Photo. The splits for the citation networks are from Yang et al. (2016). For Amazon Photo, we split nodes into train/validation/test following Shchur et al. (2018); we use 20 examples per class for training, 30 examples per class for validation, and all remaining nodes as test. See appendix A.1 for details.

**Experimental Setup:** We consider architecture depths in $\{2, 3, 4, 8, 16, 32, 64\}$ except for Pubmed and Amazon Photo where we used a maximum 32 layers to limit total training time. All architectures

---

[1]We will release the source code upon publication acceptance.
[2]Available at `https://github.com/dmlc/dgl`.

use the graph convolutional layers from Kipf & Welling (2017). For JKNet, DSGNN, and GCNII the last layer is dense with $\mathrm{softmax}$ activation. GCNII also has a dense layer at the input in order to map the input node attributes to the same dimension as the hidden layer representations which are combined via addition. For JKNet we used element-wise max pooling before the final dense layer to combine the node representations from all convolutional layers. For all architectures, we set the number of hidden units to $64$, $256$, $256$, $64$ for Cora, Citeseer, Pubmed, and Amazon Photo respectively. Optimization and hyper-parameter tuning details can be found in appendix A.3.

## 5.1 RESULTS AND ANALYSIS

**Fully observed setting:** We see in table 1 that, in this setting, the best method is GCNII for Citeseer and Amazon Photo and only marginally improved from the addition of group normalization for Cora and Pubmed. For Citeseer and Amazon Photo, GCNII outperforms the other architectures by a margin of approximately $0.2\%$ and $1.3\%$ using 32 and 4 layers respectively as compared to GCNII+GN and GCN with 8 and 2 layers. GCNII is the only method to benefit from added depth in the fully observed setting where, with the exception of Amazon Photo, performance was maximized for $8$ or more layers. Augmenting any of the architectures with pair normalization caused a reduction in test accuracy. On the other hand, the addition of group normalization resulted in a marginal improvement for some combinations of dataset and architecture such as, DSGNN vs DSGNN+GN on Pubmed and GCN vs GCN+GN on Cora. These marginal improvements, if any, offered by group normalization come at the cost of longer training times due to an increase in the number of hyper-parameter sets that must be evaluated during grid-based cross-validation (see appendix A.3 for details and discussion). Thus, the increased computation time coupled with marginal improvements suggests that there is no justification for using group normalization in the fully observed setting.

**Missing-feature setting:** Table 2 shows that all architectures benefited from increased network depth in the missing-feature setting. We note that test accuracy is lower than in the fully observed setting which indicates that input node attributes contain useful information that GNNs exploit to improve prediction accuracy. In some cases, such as Citeseer, the gap in test accuracy between the fully observed and missing-feature settings is as much as $20\%$ (GCNII+GN). Generally, all architectures except GCN achieve the highest test accuracy at a larger network depth. This is expected since information from nodes with features must propagate further in the graph to inform the representations for the nodes with missing features.

Additionally, we observe that DSGNN's ability to learn discriminative representations across all hidden layers yields benefits especially in combination with pair normalization. DSGNN enhanced with pair normalization (DSGNN+PN) was best for Amazon Photo and Pubmed by margins $2.2\%$ and $1.9\%$ from the second best architecture (GCNII+GN) respectively. DSGNN+PN was second best on Cora and Citeseer lagging by margins of $0.9\%$ and $2.7\%$ respectively from the best architecture GCNII+GN. However, as we can see in Figure 2 (right, see also appendix A.3) GCNII+GN requires approximately a factor of approximately 3 times longer to train as compared to DSGNN+PN; for practical applications with a constrained computational budget, the trade-off between a small loss in test accuracy and a large gain in training speed suggests a preference for DSGNN+PN as the architecture of choice.

**Over-smoothing resilience:** In order to examine the architectures' resilience to over-smoothing, we plot test accuracy vs network depth for a subset of the methods in fig. 2 (left) on Amazon Photo under the missing-feature setting. Here we have selected vanilla GCN as a baseline; the top performing method (DSGNN+PN) and the top performing methods in the fully observed setting (GCNII and GCNII+GN). A comprehensive set of results is given in the appendix (fig. 3 and fig. 4). We see in fig. 2 that all architectures achieve an increase in performance in step with a larger number of layers up to a point. However, we note that GCNII (which performed best on this dataset in the fully observed case), does very poorly in comparison with the other methods when using eight or more layers. This behavior is significantly improved when using it in conjunction with group normalization (GCNII+GN). In contrast with GCNII and similarly to GCNII+GN, DSGNN+PN monotonically increases performance up to sixteen layers and only slightly worsens after that.

**Additional results in appendix:** Analyzing figs. 3 and 4, we can see that all architectures with the exception of vanilla GCN alleviate over-smoothing. Generally, adding residual connections and/or some form of layer normalization is beneficial even though it does not always increase test accuracy,

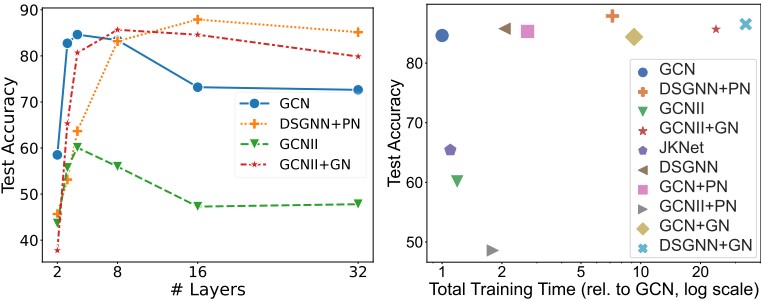

Figure 2: (Left) Plot of test accuracy vs GNN depth for a subset of the models in table 2. (Right) Plot of test accuracy vs total training time (including hyper-parameter tuning, see appendix A.3) relative to GCN. For each model, we plot the training time and accuracy for the depth where test accuracy was highest as listed in table 2. Both plots are for Amazon Photo and the missing-feature setting.

as can be seen in the fully observed setting. Similarly to the Amazon Photo dataset, in the missing-feature setting, for all architectures except vanilla GCN, there is an increase in performance in step with the increase in the number of network layers up to a point depending on the dataset. We further illustrate how all architectures except vanilla GCN are resilient to over-smoothing in figure 5 where we plot the instance information gain metric proposed by Zhou et al. (2020a) vs the number of network layers for models trained on Amazon Photo. Instance information gain measures the amount of information in the input node features that is carried through to the network's output. For architectures such as vanilla GCN that suffer from over-smoothing, this metric tends to zero as network depth increases. As we can see in Figure fig. 5, all architectures with some form of residual connections and/or normalization layers propagate information in the input node features through to the output layer as information gain is constant as a function of depth.

**Deep supervision:** Our proposed DSGNN alleviates but does not eliminate over-smoothing similarly to JKNet. This can be seen in the top row of Figure 3 where, for 3 of the 4 datasets DSGNN exhibits only a small reduction in test accuracy as a function of network depth. Furthermore, as shown in table 1, DSGNN performs competitively with the other architectures. In the missing-feature setting (see table 2), DSGNN in combination with pair normalization achieves the highest test accuracy for 2 of the 4 datasets while it is second highest for the other 2.

## 6 CONCLUSIONS & RECOMMENDATIONS

We have revisited the over-smoothing problem in deep GNNs, carrying out a systematic evaluation of several state-of-the-art solutions and their combination in the fully observed and missing-feature settings on several benchmark datasets. Additionally, we have proposed DSGNN, a new architecture based on deep supervision as a possible solution.

**Recommendations:** Based on experiments and analysis of results, we can make the following recommendations for training deep GNNs resilient to over-smoothing: (a) residual connections in any of the many forms as present in JKNet, DSGNN, and GCNII, are essential to alleviating over-smoothing in deep GNNs. Residual connections alone do not guarantee improved generalization accuracy in the fully observed setting but are essential in the missing-feature setting. (b) In the fully observed setting GCNII achieves highest test set performance and it is the only architecture that can exploit model depth. Since GCNII has the same number of trainable parameters as GCN and fewer than DSGNN and JKNet (see appendix A.3) it should be the preferred architecture. (c) In the fully observed setting, pair normalization negatively impacts performance for all architectures and its use should be avoided. (d) In the missing-feature setting, DSGNN with pair normalization and/or GCNII with group normalization should be preferred. Since GCNII and group normalization have a large number of hyper-parameters that require tuning, DSGNN with pair normalization should be preferred when computation budget is limited. We hope our analysis helps guide practitioners and researchers in their application of deep GNNs to real-world problems and promotes a more systematic and careful evaluation of existing and new GNN methods.

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

# A  APPENDIX

## A.1  DATASETS

Table 3: Dataset statistics.

| Name | Nodes | Classes | # Node features features | # Edges | Avg. Node degree | Median Node degree | # train/val/test |
|------|-------|---------|---------------------------|---------|-------------------|---------------------|------------------|
| Cora | 2708 | 7 | 1433 | 10556 | 3.9 | 3 | 140/500/1000 |
| Citeseer | 3327 | 6 | 3703 | 9228 | 2.7 | 2 | 120/500/1000 |
| Amazon Photo | 7650 | 8 | 745 | 238,163 | 31.1 | 22 | 160/240/7250 |
| Pubmed | 19717 | 3 | 500 | 88651 | 4.5 | 2 | 60/500/1000 |

Cora, Citeseer, and Pubmed are citation networks where the goal is to predict the subject of a paper. Edges represent citation relationships. The datasets have known train/val/test splits from Yang et al. (2016). Training sets are small with the number of labeled nodes equal to 140 (20 for each of 7 classes), 120 (20 for each of 6 classes), and 60 (20 for each of 3 classes) for Cora, Citeseer, and Pubmed respectively.

Amazon Photo is a subset of the Amazon Co-purchase product dataset. Nodes represent items and edges connect items that are bought together. Node attributes are bag-of-words features derived from product reviews. The goal is to predict the product category.

We treat all graphs as undirected as it is common in the GNN literature.

## A.2  ADDITIONAL RESULTS

We demonstrate the architectures' resilience to over-smoothing in the fully observed and missing-feature settings by plotting test accuracy vs network depth in Figures 3 and 4.

In the fully observed setting, the performance of GCN drops sharply for all datasets for deep models. The combination of GCN with pair (GCN+PN) or group (GCN+GN) normalization alleviates the problem to some degree but performance still degrades rapidly with depth (middle and bottom rows of Figure 3). GCN+GN is more resilient to over-smoothing than GCN+PN. GCNII and JKNet both alleviate over-smoothing. DSGNN is better than GCN, GCN+PN, and GCN+GN but still only reduces the rate of over-smoothing.

In the missing-feature setting, all architectures initially benefit from increased depth as more graph convolutional layers are necessary to allow for information to propagate through the graph given that the majority of nodes do not have associated input node attributes (which have been set to zero vectors). However, GCN still suffers from over-smoothing as network depth increases while the other architectures are more resilient. Some combinations such as DSGNN+PN, DSGNN+GN, and GCNII+GN on Pubmed continue to see an increase on test accuracy for up to 32 layers.

## A.3  TRAINING DETAILS

We used the Adam optimizer with learning rate set to 0.01. We trained all models for a maximum 2500 epochs and used early stopping with patience 250 epochs monitoring validation set accuracy. We run each experiment 20 times and report the mean performance and standard deviation of test accuracy.

The training time for each architecture depends on two main factors. One is the total number of trainable parameters and the other is the number of hyper-parameters that require tuning. In table 4, we report for each architecture the total number of hyper-parameters, hyper-parameter settings considered using grid search and the total number of trainable parameters.

For all architectures we tune dropout (4 values from $\{0.2, 0.5, 0.6, 0.7\}$) and weight decay (5 values from $\{0.01, 0.001, 0.0001, 0.0005, 0.00001\}$). Weight decay for the dense layers in GCNII was fixed to 0.0005 as suggested by Chen et al. (2020b). GCNII requires selection of one additional

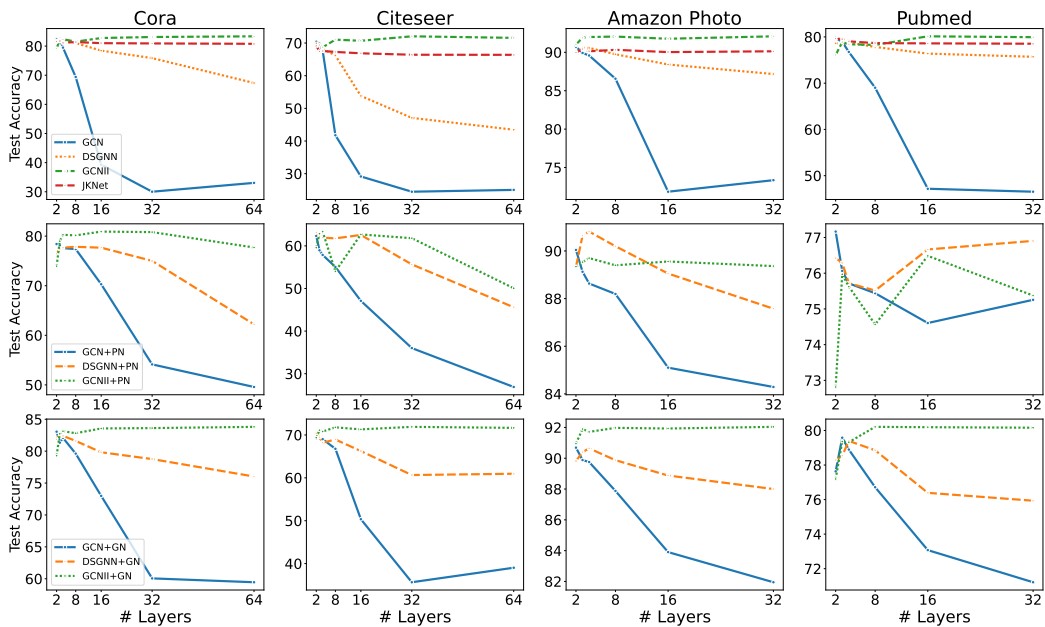

Figure 3: Test accuracy vs architecture depth for the **fully observed setting**. Top row: vanilla architectures. Middle row: architectures with pair normalization (+PN). Bottom row: architectures with group normalization (+GN). Note that the y-axis for the different plots have different range.

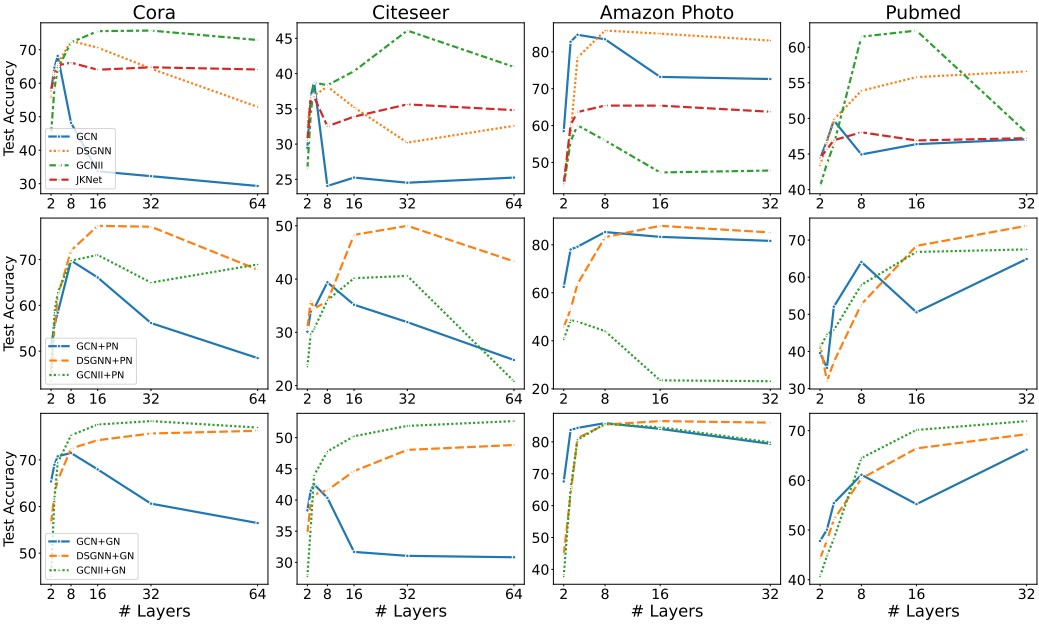

Figure 4: Test accuracy vs architecture depth for the **missing-feature setting**. Top row: vanilla architectures. Middle row: architectures with pair normalization (+PN). Bottom row: architectures with group normalization (+GN). Note that the y-axis for the different plots have different range.

hyper-parameter, the identity mapping weight, (3 values from $\{0.4, 0.5, 0.6\}$). For GCNII, we fixed the value indicating the fraction of input features retained at each layer to $0.1$ as in Chen et al. (2020b). Pair normalization layers do not add any hyper-parameters. Group normalization layers require selection of an additional hyper-parameter, the group normalization balancing factor, (9

Table 4: Number of hyper-parameter sets and number of trainable parameters for each architecture.

| Architecture | # hyper-parameters | # hyper-parameter settings | # of trainable parameters |
| --- | --- | --- | --- |
| GCN | 2 | 20 | $(K-1) \times d^2 + d \times C$ |
| JKNet | 2 | 20 | $K \times d^2 + d \times C$ |
| DSGNN | 2 | 20 | $K \times d \times (d+C)$ |
| GCNII | 3 | 60 | $(K+1) \times d^2 + d \times C$ |
| GCN+PN | 2 | 20 | $(K-1) \times d^2 + d \times C$ |
| DSGNN+PN | 2 | 20 | $K \times d \times (d+C)$ |
| GCNII+PN | 3 | 60 | $(K+1) \times d^2 + d \times C$ |
| GCN+GN | 3 | 180 | $(K-1) \times d^2 + d \times (C + K \times G)$ |
| DSGNN+GN | 3 | 180 | $K \times d \times (d+C+G)$ |
| GCNII+GN | 4 | 540 | $(K+1) \times d^2 + d \times (C + K \times G)$ |

values from $\{0.0005, 0.001, 0.002, 0.003, 0.005, 0.01, 0.02, 0.03, 0.05\}$). For group normalization we set the number of groups to $10, 10, 5$ for Cora, Citeseer, and Pubmed respectively as suggested by Zhou et al. (2020b). For Amazon Photo, we set the number of groups to 10 since Zhou et al. (2020b) recommends that the number of groups should be close to the number of classes to predict.

The total number of hyper-parameters that require tuning varies based on the architecture. For example, the vanilla GCN architecture only requires tuning over a set of $4 \times 5 = 20$ hyper-parameter settings whereas GCNII+GN requires tuning over a set of $4 \times 5 \times 3 \times 9 = 540$ hyper-parameter settings.

We estimate the total number of trainable parameters for each architecture and report them in table 4. Let $K$ be the number of GNN layers, $C$ the number of classes, $G$ the number of node clusters, and $d$ the dimensionality of latent node representations. For clarity, we assume that nodes have input attributes also with $d$ dimensions.

In our experimental setup, a $K$-layer GCN model comprises of $K$ graph convolutional layers where the last layer is considered the output layer predicting class probabilities for each node. For JKNet and GCNII, a $K$-layer model comprises of $K$ graph convolutional layers followed by a dense layer for classification. Figure 1 (left) shows an example of JKNet with $K = 2$ layers where 2 are graph convolutional. For DSGNN, a $K$-layer model comprises of $K$ graph convolutional layers and $K$ dense layers attached to each hidden layer. Figure 1 (right) shows an example of DSGNN with $K = 2$ layers where 2 are graph convolutional and 2 are dense with $\mathrm{softmax}$ activation.

The number of trainable parameters for GCN are $(K-1) \times d^2 + d \times C$. Here we consider the last graph convolutional layer as the output layer. JKNet has $K$ graph convolutional layers followed by a dense layer. The total number of trainable parameters is $K \times d^2 + d \times C$. DSGNN adds a dense layer for each graph convolutional layer so the number of trainable parameters is $K \times d^2 + K \times d \times C$. GCNII adds a dense layer at the input followed by $K$ graph convolutional layers and a dense layer for classification. Hence, the total number of trainable parameters is $(K+1) \times d^2 + d \times C$. Group normalization adds a dense layer with $\mathrm{softmax}$ activation for each graph convolutional layer; the dense layer calculates a soft assignment of nodes to clusters. For a model with $K$ graph convolutional layers, group normalization adds $K \times d \times G$ trainable parameters.

### A.4 QUANTIFYING OVER-SMOOTHING

We can quantify the propensity of an architecture to over-smooth by considering the amount of information in the input node features that is not preserved at the output as proposed by Zhou et al. (2020b). Instance information gain measures the amount of information in the input node features that is preserved in the output of a GNN layer. Zhou et al. (2020b) propose instance information gain goes to zero as a function of GNN depth indicating over-smoothing. In Figure 5, we plot instance information gain as a function of architecture depth for Amazon Photo in the fully observed setting. We see that information gain for GCN decreases as network depth increases. This is not the case for

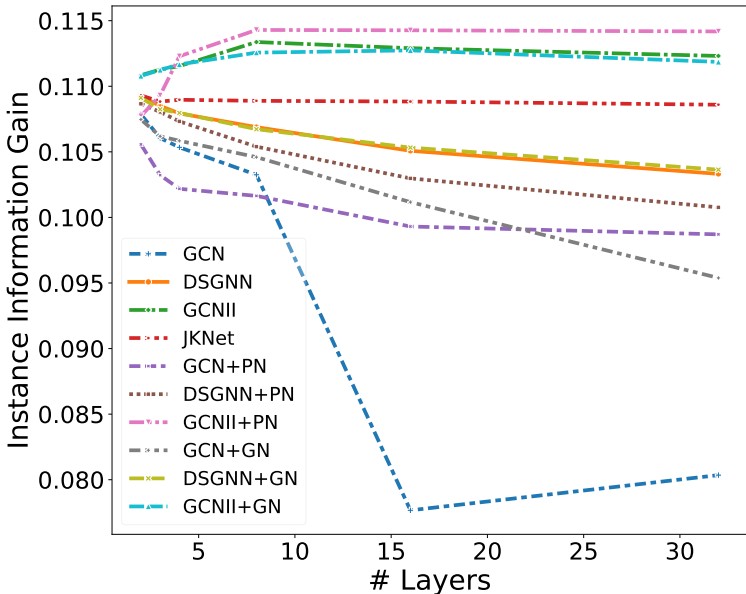

Figure 5: Instance Information Gain as a function of GNN depth. All results are for the Amazon Photo dataset in the fully observed setting.

all other architectures. Notably, just the addition of pair normalization to GCN, GCN+PN, alleviates information loss. This results correlates with the test accuracy vs GNN depth curves in Figure 3.

