# OpenReview forum: "Revisiting Over-smoothing in Graph Neural Networks"
_ICLR.cc/2023/Conference — Submitted to ICLR 2023_

### Official Review · Reviewer_QdpM · 2022-10-22

**Confidence:** 4
**Correctness:** 3
**Technical Novelty And Significance:** 2
**Empirical Novelty And Significance:** 2
**Recommendation:** 3

**Clarity, Quality, Novelty And Reproducibility:**

The paper is easy to follow. The paper is in general of high quality. The novelty is uncertain since benchmarking deep GNNs has already been proposed in previous work. I trust that the results can be reproduced.

**Details Of Ethics Concerns:**

No concern.

**Strength And Weaknesses:**

Strengths:
1. Over-smoothing is an important problem for GNNs.
2. The idea of deeply supervised GNNs (DSGNN) is well-motivated. The performance of DSGNN improves in most cases as the number of layers increases.

Weaknesses:
1. The benchmarking experiments are not enough. Particularly, there is a previous work that conducts much more comprehensive benchmarking than this paper [1].
2. Only 4 datasets are considered in the evaluation and benchmarking. More large-scale datasets should be considered.
3. DSGNN seems to only be comparable with GCNII. It is unclear what advantages DSGNN has.
4. A missing related work should be compared since it can also improve performance when using deeper architectures [2].


[1] Chen, Tianlong, et al. "Bag of tricks for training deeper graph neural networks: A comprehensive benchmark study." IEEE Transactions on Pattern Analysis and Machine Intelligence (2022).
[2] Zhou, Kaixiong, et al. "Dirichlet energy constrained learning for deep graph neural networks." Advances in Neural Information Processing Systems 34 (2021): 21834-21846.

**Summary Of The Paper:**

This paper studies the over-smoothing problem in GNN, which means the node representations tend to become indistinguishable as the number of layers increases. The key contributions of these papers are two-folded. First, they do a benchmarking to evaluate and analyze several methods. Second, motivated by deeply supervised nets, the paper proposed deeply supervised GNNs. Experimental results on standard and missing feature scenarios show the method's advantages.

**Summary Of The Review:**

The paper tackles an important problem. However, there is a very similar related work that also focuses on benchmarking deep GNNs. The previous work considers more datasets and more scenarios than this work. So it is unclear to me whether the contribution of this paper is significant. The paper should also consider more datasets and baselines. Thus, overall I can not recommend acceptance.

---

### Official Review · Reviewer_K9Pi · 2022-10-23

**Confidence:** 4
**Correctness:** 2
**Technical Novelty And Significance:** 2
**Empirical Novelty And Significance:** 2
**Recommendation:** 3

**Clarity, Quality, Novelty And Reproducibility:**

- *Clarity*: this paper is clearly written.
- *Quality*: the quality of the experiment is not good. As mentioned above, a lot of baselines are not considered and the experiments cannot justify their claims.
- *Novelty*: this paper lacks of novelty. Using deep supervision to improve the performance of deep networks has been proposed and studied.
- * Reproducibility*: the code is not provided, but the authors provide hyperparamters in Apeendix A.3. Given that the prepose method can be easily implemented, I believe that it can be reproduced.

**Strength And Weaknesses:**

### Strengths
- The paper is written clearly and easy to follow. I like section 3 which summarizes the prior efforts on tackling over-smoothing in a clear manner.
- The proposed method is simple.
- This paper studies both fully observed and missing feature settings.

### Weaknesses
- No evidence is provide to support this paper's claim that DSGNN can address over-smoothing. It is well known that deep supervision helps the training of deep nets. It is not clear whether deep supervision actually addresses the over-smoothing problem. There are two other potential explanations of the good: 1) deep supervision alleviates the gradient vanishing problem of the bottom layers and 2) the model makes predictions by taking the average which is an ensemble method that usually improves the performance of models.
- The author should report the performance of the last classifier of DSGNN. It is possible that the top layers are as bad as deep GCNs and DSGNN does well relying on the predictions of its earlier layers.
- The author should consider a simple baseline that averages a couple of different shallow GCNs which also has the advantage of ensemble. For example, comparing a 32 layer DSGNN with these baselines
   - 16 2-layer GCNs
   - 4 2-layer GCNs + 4 3-layer GCNs + 4 4-layer GCNs
   - a combination of 1, 2, 3, ..., 32 layer GCNs
- The authors only consider architecture depths in {2, 3, 4, 8, 16, 32, 64}. However, as we can see in Table 2, several models have the best depth being 64. It is weird that the author didn't consider increasing the depth further to find the optimal depth.

**Summary Of The Paper:**

This paper studies the over-smoothing problem in graph neural networks and proposes a new architecture called deeply-supervised GNN (DSGNN) as a solution. Inspired by deeply-supervised nets (Lee et al., 2015), DSGNN has a classifier attached to each hidden layer and gets supervision during training. During inference, DSGNN makes predictions based on averaged prediction across all layers. The authors conduct experiments on Cora, Citeseer, Amazon Photo, and Pubmed datasets with both fully observed and missing feature settings to compare GCN, JKNet, GCNII, and their proposed DSGNN. The application of pair normalization and group normalization are also considered in the experiments.

**Summary Of The Review:**

To sum up, the main claim of the paper is not well-supported and a lot of baselines are missing. Therefore, I recommend rejecting this paper.

---

### Official Review · Reviewer_FUiX · 2022-10-24

**Confidence:** 5
**Correctness:** 3
**Technical Novelty And Significance:** 1
**Empirical Novelty And Significance:** 1
**Recommendation:** 3

**Clarity, Quality, Novelty And Reproducibility:**

+ The overall quality of this work is below the average bar of ICLR papers;
+ The clarity is generally good since the presentation and details are easy to follow;
+ The originality of this work is not strong, see the weaknesses mentioned above.

**Strength And Weaknesses:**

Strengths:

+ The discussion on the characteristics of  several existing GNN over-smoothing solutions and their combinations is informative
+ The presentation and readability are easy to follow

Weaknesses:
+ The technical contribution is limited since the deeply-supervised idea is already a widely-used trick for designing very deep nets, like DenseNet (CVPR, 2017) and its follow-up works
+ More in-depth (theoretical) analysis on why the proposed DSGNN is powerful is missing. The arguments presented in Section 4.2 are not convincing.
+ Experimental studies are not strong. More large-scale graph datasets are needed to further verify the effectiveness and efficiency of DSGNN. Also, it seems that GCNII+GN outperforms DSGNN in most cases, which contradicts the merits of DSGNN.

**Summary Of The Paper:**

This paper tries to study the over-smoothing issue of GNNs. A detailed discussion on the potential 'pitfalls' of the existing works (focusing on preventing over-smoothing) is given. Then, a deeply-supervised GNN framework, based on the idea of layer-wise supervision, is proposed for problem-solving. Extensive experiments are conducted to verify the effectiveness and advantage of the proposed deeply-supervised trick. Based on the experimental results, some (initial) recommendations for the selection of a GNN architecture for practical applications are presented.

**Summary Of The Review:**

Overall, based on the weaknesses mentioned above, I vote for rejection.

---

### Official Review · Reviewer_NBPk · 2022-10-24

**Confidence:** 4
**Correctness:** 3
**Technical Novelty And Significance:** 1
**Empirical Novelty And Significance:** 2
**Recommendation:** 3

**Clarity, Quality, Novelty And Reproducibility:**

This work has a clear presentation which is an advantage. However, form perspective of technique and analysis, the quality of this work can be further improved. The method of this work is somewhat novel. However, without in-depth analysis and adquate evidence, I currently think the prosoed method is weak. Finally, the reproducibility of this last article is moderate.

**Strength And Weaknesses:**

The main strength of this work is the clear writing. It is not hard to understand the main contribution of this work. Overall I like the presentation of this work. I believe that the authors make an effort in polishing the writing.

On the other hand, I think that the experimental results presented in this work are systematic. The authors select the most famous deep GNN method and make a detailed comparison between them.

However this work lacks in-depth analysis on the difference between previous methods and the proposed method.

First, the authors claim that previous approaches may have potential pitfalls because of the evaluation and fairness problem. I understand the author may try to emphasise the advantage of the proposed method in missing features scenarios. However, the discussion in Section 3.2 is not convincing. The author only explains the possible risks of the previous method, and does not give us a specific and in-depth analysis to tell you why the previous method has shortcomings and the reasons for the shortcomings.

Second, when expounding the advantages of deep supervision, the author does not provide effective and detailed arguments to be convincing. What I mean by effective arguments include theoretical proofs or elaborate experimental analyses. However I don't see any analysis. Therefore, the lack of this paper is able to show the proposed method well.

Finally, from the reported results in Table 1 and 2, the best performance usually is achieved by GCNII+GN instead of the proposed method (DSGNN). Even in the missing features setting, the proposed method of this work only achieves best performance on 2 out of 4 dataset. This is not enough to demonstrate the effectiveness of the method proposed in the article.


**Summary Of The Paper:**

This work studies the over-smoothing problem in deep graph neural networks. The authors systematically evaluate different methods including graph convolutional networks, jumping knowledge networks, pair normalization, group normaliztion, and GCNII. Besides, the authors propose an architecture called architecture deeply supervised GNNs (DSGNNs) based on the idea of layer-wide supervision. The experimental results show that the proposed method can achieve better performance on deeper GNNs, especially providing best performance in missing feature scenarios.


**Summary Of The Review:**

Although this paper is complete in content, the fatal problem is that there is no effective analysis to demonstrate the superiority of the method. I suggest that the author think clearly about the advantages of the method in this work, and reorganize the presentation of the article based on this. Based on the current version, I don't think it meets the ICLR requirements for reception. Authors are encouraged to further improve this article from originality and technicality.

---

### Author Response · Authors · 2022-11-18
**Our response for all reviewers**

We thank the reviewers for their constructive feedback. While we disagree with some of their comments with regards to experimental evaluation and analysis of our approach, we appreciate their time in pointing out how we can improve our paper by, for example, highlighting differences (and potentially comparing) with related (although contemporaneous) related work.

---

### Decision · Program_Chairs · 2023-01-20

**Decision:**

Reject

**Justification For Why Not Higher Score:**

Limited technical novelty.

**Justification For Why Not Lower Score:**

N/A

**Metareview: Summary, Strengths And Weaknesses:**

This paper studies the over-smoothing problem in GNN, presents a detailed discussion on the potential pitfalls of existing methods and proposes a deeply-supervised GNN (DSGNN) architecture as a solution. Empirical evaluation shows some advantage of the proposed DSGNN architecture . Based on the experimental results, some (initial) recommendations for the selection of a GNN architecture for practical applications are presented.
Strengths:
Well written paper with informative discussion on the characteristics of existing GNN over-smoothing solutions

Weaknesses:
Limited technical contribution as the main idea has been widely used for very deep networks.
Lacking in-depth (theoretical) analysis.
Experimental results are not strong.